

# Seasonal origins of soil water used by trees

Scott T. Allen[1,2*], James W. Kirchner[1,3,4], Sabine Braun[5], Rolf T.W. Siegwolf[2,3], Gregory R. Goldsmith[2,6]

[1]Department of Environmental Systems Science, ETH Zurich, Zurich, 8092, Switzerland.
[2]Ecosystem Fluxes Group, Laboratory for Atmospheric Chemistry, Paul Scherrer Institute, Villigen, 5232, Switzerland.
[3]Swiss Federal Research Institute WSL, Birmensdorf, 8903, Switzerland
[4]Department of Earth and Planetary Science, University of California, Berkeley, California, USA
[5]Institute for Applied Plant Biology, Witterswil, 4108, Switzerland
[6]Schmid College of Science and Technology, Chapman University, Orange CA, 92866, USA

*Correspondence to*: Scott T. Allen (allensc@ethz.ch)

**Abstract.** Rain recharges soil water storages and either percolates downward into aquifers and streams, or is returned to the atmosphere through evapotranspiration. Although it is commonly assumed that summer rainfall recharges plant-available water during the growing season, the seasonal origins of water used by plants have not been systematically explored. We characterize the seasonal origins of waters in soils and trees by comparing their mid-summer isotopic signatures ($\delta^2$H) to seasonal isotopic cycles in precipitation, using a new seasonal origin index. Across 182 Swiss forest sites, xylem water isotopic signatures show that summer rain was not the predominant water source for mid-summer transpiration in any of the three sampled tree species. Beech and oak mostly used winter precipitation, whereas spruce used water of more diverse seasonal origins. Even in the same plots, beech consistently used more winter precipitation than spruce, demonstrating consistent niche partitioning in the rhizosphere. All three species' xylem water isotopes indicate that trees used more winter precipitation in drier regions, potentially mitigating their vulnerability to summer droughts. The widespread occurrence of winter isotopic signatures in mid-summer xylem implies that growing-season rainfall may have minimally recharged the soil water storages that supply tree growth, even across diverse humid climates (690-2068-mm annual precipitation). Beyond these ecological and hydrological implications, our findings also imply that stable isotopes of $\delta^{18}$O and $\delta^2$H in plant tissues, which are often used in climate reconstructions, may not reflect water from growing-season climates. More broadly, these results conflict with common assumptions on tree water use and provide empirical support for developing more realistic concepts of how water flows through soils and is accessed by roots.



## 1 Introduction

Plant water availability shapes ecosystems, climates, and natural resources. In hydrology and ecology, soil water storage is often represented as a bucket or vertical stack of well-mixed reservoirs, filled by previous precipitation events, and used by plants as a function of their rooting depth (Lawrence, et al., 2011; Wigmosta et al., 1994). The reality is more complex;

water transport through soils tends to be dominated by preferential flow through large pores, whereas water is often primarily stored in the finer matrix (Beven and Germann, 1982; Lawes et al., 1882). Thus plant water availability depends on the interplay between macropore flow, matrix storage, and the rooting architecture of vegetation (Brooks et al., 2010; Stewart et al., 1999; Tinker, 1976). Previous research has documented depths of roots and root water uptake (Fan et al., 2017; West et al., 2012), but little attention has been directed towards understanding how water becomes available for uptake at those depths.

Water stable isotope signatures ($\delta^2$H and $\delta^{18}$O) have been used as tracers to show that plant water uptake is not sourced from the same subsurface storage as streamflow (Evaristo et al., 2015; Good et al., 2015; Javaux et al., 2016), but it remains unclear how that storage is replenished and becomes available to plants. Soils may retain a mixture of waters that originate from many previous precipitation events (Mueller et al., 2014; Sprenger et al., 2016), but plants may not evenly sample from that distribution of water ages, because plants may root such that they preferentially take up water moving along faster or

slower pathways (Brooks et al., 2010; Ehleringer et al., 1991; Stewart et al., 1999). These interactions between root distributions and infiltration dynamics could hypothetically result in plants disproportionally using precipitation from past seasons, rather than recent precipitation. While a few case studies have reported plants predominantly using precipitation from past seasons in arid (Ehleringer et al., 1991) or Mediterranean climates (Brooks et al., 2010; Rempe and Dietrich, 2018) where there is minimal growing-season precipitation, the seasonal origins of water used by plants have not been systematically

explored in humid climates.

To investigate the seasonal origins of waters that supply mid-summer tree growth, we analyzed xylem water isotopes from a snapshot sample of 918 trees from three dominant species in 182 forest sites across Switzerland. At 31 of these sites, we complemented the xylem water with isotope values of soil waters, sampled using suction lysimeters (which access the more mobile fraction of soil water). To characterize the seasonal origins of xylem water and lysimeter soil water, we developed a

seasonal origin index, based on the isotopic signature of soil and plant water relative to seasonal precipitation isotope cycle; this index quantifies the over-expression of winter versus summer (recent) precipitation in xylem or lysimeter waters, relative to annual precipitation. This new seasonal origin index can be effectively used in these sites because the strong seasonal isotopic cycle in Swiss precipitation (Allen et al., 2018) allows for winter and summer precipitation to be clearly distinguished in tree xylem. We used this mid-summer snapshot to determine a) whether summer or winter precipitation was over-

represented in mid-summer soil and xylem waters, relative to annual precipitation, b) how the seasonal origins of xylem water varied across diverse climates and site characteristics, and c) whether these three dominant trees species differed in their water sources.



## 2 Materials and Methods

### 2.1 The seasonal origin index

To characterize when xylem water and lysimeter soil water originated as precipitation, we developed a seasonal origin index (SOI),

$$\text{SOI}= \begin{cases} \frac{\delta_x - \delta_{annP}}{\delta_{summerP} - \delta_{annP}}, \text{ if } \delta_x > \delta_{annP} \\ \frac{\delta_x - \delta_{annP}}{\delta_{annP} - \delta_{winterP}}, \text{ if } \delta_x < \delta_{annP} \end{cases} \quad (1),$$

where $\delta_x$ is the fractionation-compensated $\delta^2$H isotopic signature of xylem water or lysimeter soil water, and $\delta_{winterP}$, $\delta_{summerP}$, and $\delta_{annP}$ are the $\delta^2$H isotopic signatures of typical winter, typical summer, and volume-weighted annual precipitation at each study site (see sect. 2.4 and Fig. 1). This index expresses the isotopic signature of soil and plant water relative to seasonal precipitation isotope cycles, which are especially strong in high-latitude, continental interiors, where precipitation isotopes are heavy in summer and light in winter (Halder et al., 2015; Vachon et al., 2007). The SOI will be near -1.0 for soil and plant water samples derived entirely from winter precipitation, and near 1.0 for samples derived entirely from summer precipitation (Fig. 1). Samples with SOI values near zero approximate the annual average precipitation, and can potentially represent many possible mixtures of waters from spring, summer, autumn, and winter. By extracting waters from tree xylem, which reflect the waters taken up by roots (Newberry et al., 2017), and comparing those data to precipitation isotopes, this metric is robust to several uncertainties that are prevalent in isotope-based rooting depth studies, such as sampling and extracting soil waters that are representative of the waters accessed by roots, as described below in greater detail (Goldsmith et al., 2018; Orlowski et al., 2018; Penna et al., 2018).

In using this SOI, we implicitly test the null hypothesis that xylem and lysimeter soil water are the annual volume-weighted average of precipitation, and thus we center the SOI index such that SOI=0 at that value, in any precipitation regime (Fig. 1). We address the question: "is winter or summer water over-represented in soils or xylem, relative to volume-weighted precipitation?". Importantly, this SOI equation (eq. 1) differs from a simple, two-end member ($\delta_{winterP}$ and $\delta_{summerP}$) mixing model, which addresses a different question – "is there more winter water than summer water in soils or xylem?" – but does not account for the fact that we should expect more winter precipitation in soils (for example) at sites with more winter rainfall. Thus the piecewise linear equation that we use to define SOI is more appropriate for determining whether winter or summer water is over-represented in soils and xylem (relative to precipitation inputs) across sites with different seasonal patterns in precipitation; nonetheless, the two approaches yield similar values in areas with relatively even precipitation throughout the year, such as Switzerland (Figure S1).

### 2.2 Field sites and measurements

The study was carried out in summer of 2015 at 182 sites established across Switzerland as part of a forest health monitoring program (Braun et al., 1999, 2017). Each site contained at least one of three tree species: 97 contained beech (*Fagus*



*sylvatica* L.)*,* 71 contained spruce (*Picea abies* (L.) H. Karst.)*,* and 49 contained oak (*Quercus robur* L.). Sites ranged from 255 to 1840 m a.s.l. in elevation, 3.3 to 11.1 °C in mean annual temperature, and 690 to 2068 mm in total annual precipitation. The mean elevations of sites with oak (513 m a.s.l.) and beech (617 m a.s.l.) were slightly lower than those of sites with spruce (893 m a.s.l.). On average, mean annual precipitation at sites with oak (1085 mm yr$^{-1}$) was slightly lower than at sites with

beech (1285 mm yr$^{-1}$) or spruce (1339 mm yr$^{-1}$). Tree diameters, measured in 2014, ranged from 17 to 105 cm. All stands are actively managed and composed of mature trees in established, closed-canopy stands. Soils are highly variable in depth (ranging from 30 to 220 cm) and texture (ranging from 4 to 61% clays and 6 to 81% sands in the top 50 cm) across the sites (see Figs. S2 and S3).

To determine the δ$^{18}$O and δ$^{2}$H ratios of xylem water in trees, branches were sampled from 3-8 individual trees of

each of the species present in each plot. All branch samples were collected between 27 July and 10 August 2015, using pole pruners operated by technicians suspended below helicopters. Thus, all sampled trees occupied at least intermediate canopy positions. On the ground, bark and phloem tissue was removed from fully suberized branches and samples were sealed in vials and frozen for later extraction and isotopic analysis.

To determine the δ$^{18}$O and δ$^{2}$H ratios of mobile soil waters, samples were collected from suction lysimeters

(Soilmoisture Equipment Corp., Santa Barbara, USA) at 31 of the forest monitoring sites in July 2015. A tension of 60-70 kPa was applied to each lysimeter once, within one month prior to the xylem sample collection date. Water samples were collected from lysimeters where water could be extracted (i.e., a tension could be applied without losing suction), approximately four to five weeks after the tension was applied. Each site had sets of lysimeters at one to four different depths (up to 120 cm deep) depending on the site's soil depth. For each depth and at each of the 31 sites, there were three to eight replicate lysimeter sets

(mean of 6.7). Replicate samples were pooled by depth, then sealed and frozen in 50 ml vials for later isotopic analysis. We used these lysimeter data to understand the seasonal origins of the mobile fractions of water in soils (Brooks et al., 2010; Sprenger et al., 2016a), and do not assume that they are representative of the entire soil-water pool, or the pool of water available to plants.

To determine how the δ$^{18}$O and δ$^{2}$H ratios of xylem water and lysimeter soil water varied as a function of soils and

climate, additional site metrics were measured. Soil texture (sand, silt, clay, stone, and organic matter content) and root density by horizon were determined from a soil pit excavated at each site to characterize soil properties. Measurement protocols were consistent with the German soil survey (German BGR, 2005). Elevation was determined for each site from a digital elevation model (25 m resolution; Swiss Federal Office of Topography, Wabern, Switzerland). Slope and aspect were surveyed at each field site, using a compass and clinometer. Mean temperature, precipitation amount, and potential evapotranspiration (PET)

were determined using a geospatial model (Meteotest, Bern, Switzerland) based on weather station data. To interpret the root density indices assigned to each soil horizon, we converted the ordinal density indices assigned in the field soil survey (W1, W2, W3, W4, W5, W6; German BGR, 2005) to the respective mean values of the index categories (1.5, 4, 8, 15.5, 35.5, and



50 roots cm$^{-2}$); the density-weighted mean root depth (i.e., depth to center of mass) was then calculated using those values for each site.

## 2.3 Sample processing and laboratory analyses

Water was extracted from branch xylem material by cryogenic vacuum distillation (West et al., 2006) at the Paul Scherrer Institute (Villigen, Switzerland). All samples were heated for 2 hours to ensure complete extraction. The δ$^{18}$O and δ$^2$H ratios of soil and xylem water were subsequently analyzed using a high temperature conversion/elemental analyzer (TC/EA) connected to a Delta Plus XP isotope ratio mass spectrometer via a Conflo III interface (Thermo Fisher Scientific, Bremen Germany). Isotope ratios are expressed in per mil (‰) notation relative to V-SMOW. The long-term instrument precision, measured using independent quality control standards, is ≤ 0.4 ‰ for δ$^2$H and ≤ 0.2 ‰ for δ$^{18}$O. There has been considerable debate over cryogenic vacuum extraction because studies have observed discrepancies in cryogenic extraction of soil water in rehydration experiments (Meißner et al., 2014; Oerter et al., 2014; Orlowski et al., 2018); however, soil waters were not cryogenically extracted in this study, and those discrepancies are not observed when extracting xylem water (Newberry et al., 2017). Physicochemical fractionation processes can occur prior to sampling within plants or at the soil-root interface in xerophytic or halophytic woody plants (Ellsworth and Williams, 2007; Zhao et al., 2016), or at the time of initial leaf flush (Treydte et al., 2014), but the effects of those processes are irrelevant to our mid-summer sampling in a humid climate region.

## 2.4 Data processing and application in Seasonal Origin Index analysis

To compare tree xylem water with precipitation inputs at the respective sites, seasonal cycles of the δ$^{18}$O and δ$^2$H of precipitation were modeled for each site using a sinusoidal isoscape approach (Allen et al., 2018). Monthly precipitation isotope data were downloaded from 31 monitoring locations in Switzerland (NAQUA network), Austria (Austrian Network of Isotopes in Precipitation), and Germany (Global Network of Isotopes in Precipitation); 13 were in Switzerland, and the remaining 18 were within 135 km of the Swiss border. Sine functions were fitted to precipitation stable isotope measurements at each monitoring site using all available data from 2007 through 2015. Parameters describing the δ$^{18}$O and δ$^2$H sine functions (*offset*, *amplitude*, *phase*) were then interpolated across Switzerland by multiple linear regression models using site latitude, longitude, elevation, mean annual temperature range, and mean total precipitation amount as the predictors. This yielded a measure of central tendency (*offset*) and strength of seasonal cycle (*amplitude*) for δ$^{18}$O and δ$^2$H. For each site, we calculated a typical winter precipitation value ($\delta_{winterP} = offset - amplitude$), and a typical summer precipitation value ($\delta_{summerP} = offset + amplitude$). The widths of the shaded areas in Figs. 2 and S5 show $\delta_{winterP}$ and $\delta_{summerP}$ ± 2×RMSE, where RMSE is the root mean square error of predicted versus fitted *amplitude* at the precipitation isotope monitoring sites.

To calculate $\delta_{annP}$ (eq. 1), we used the volume-weighted mean precipitation of a 24-month period prior to the xylem water field sampling campaign (August 2013-July 2015). Monthly values of precipitation δ$^{18}$O and δ$^2$H at each field site were





calculated by individual-month multiple linear regression models, fitted to monthly precipitation isotope measurements at the 31 precipitation isotope monitoring sites, using site latitude, longitude, elevation, mean annual temperature range, and mean total precipitation amount as predictors (Allen et al., 2018). In an additional step to account for variations not captured by the regression model, we kriged the prediction residuals of the regression model at each precipitation monitoring station, to create

monthly adjustment layers that were added to the regression predictions (Allen et al., 2018). The mean absolute error in predicting monthly precipitation was 1.2 ‰ $\delta^{18}$O and 9.6 ‰ $\delta^2$H. Lastly, the site-specific precipitation isotope values were weighted by site-specific monthly precipitation amounts (Meteotest, Bern, Switzerland) and summed for the 24 months prior to sampling.

To calculate the Seasonal Origin Index (SOI), we used fractionation-compensated lysimeter- and xylem-water isotope

values as approximations of their source values prior to any evaporative fractionation. Deviations of soil or xylem water isotope values from local meteoric water lines (LMWLs) were treated as fractionation effects. To compensate for these fractionation effects, soil or xylem isotope values were adjusted back to their respective LMWLs along an evaporation line slope, calculated using the Craig-Gordon model as implemented for diffusion-controlled soil evaporation scenarios by Benettin et al. (2018). Two recent studies have shown that such theory-based approaches are more robust than evaporation lines fitted to soil-water

observations, which will typically be confounded by isotopic variations in precipitation over time (Benettin et al., 2018; Bowen et al., 2018). Here, we computed the evaporation line at each site as a function of summer mean relative humidity and temperature (Fig. S5). Slopes were calculated using both daily maximum temperatures and daily minimum temperatures to understand the uncertainty associated with the range of conditions under which evaporation occurs; for fractionation compensation, we used the mean of the minimum and maximum temperature slopes. Calculated fractionation slopes across

the Swiss sites were between 2.7 and 3.4 (see Fig. S5), consistent with those reported in a previous synthesis (Sprenger et al., 2016a). To compensate for the fractionation effects in xylem water and lysimeter soil water, the monthly precipitation $\delta^{18}$O and $\delta^2$H calculated above for each site were fitted by orthogonal least squares to generate site-specific LMWLs. Then, lysimeter- and xylem-water values were compensated for evaporative fractionation by shifting them along the site-specific evaporation slopes estimated above until they intersected the LMWLs. These intersection points are the fractionation-

compensated values used to represent the precipitation sources of waters in trees and soils, (i.e., they were used as the $\delta_x$ inputs for the calculation of SOI by eq. 1). The effects of this fractionation-compensation step can be seen by comparing the patterns and species differences in Figs. 2 and S5, or Figs. 5 and S6.

**2.5 Statistical Analyses**

To determine how the $\delta^{18}$O and $\delta^2$H ratios of water in soil and plants varied with soil properties, topography and

climate, we calculated Pearson and Spearman (rank) correlation coefficients between site characteristics and both xylem SOI and lysimeter SOI. Site-mean SOI was calculated for each species and for shallow ($\leq$ 30 cm) and deep (> 30 cm) lysimeters. To account for the influence of spatial clustering in correlations, the influence of any given point was weighted by $N^{-1}$, where



*N*, calculated by site and by species, is the number of sites within 10 km that contain the same species; *N* was also calculated for lysimeters. Details on the site characteristics that we examined are provided in Table S1. A stepwise multiple regression test was also performed to consider interactions between these terms (Table S2). All statistical tests were performed in MATLAB (Mathworks, Inc., Natick MA, USA).

## 3. Results and Discussion

Tree use of winter-sourced water in mid-summer was widespread among our 182 Swiss forest sites (Fig. 2). Low (i.e., winter) SOI values were also markedly more prevalent in xylem water than in lysimeter samples of soil water (Fig. 2). Overall, summer (recent) precipitation signatures were uncommon in these mid-summer samples; SOI was > 0.5 for only 1% of oak and beech samples, 5% of spruce samples, and 5% of lysimeter soil water samples (Fig. 3). Winter precipitation signatures were distinctly more common, particularly in the broadleaf trees (oak and beech); SOI was < -0.5 for 78% of oak and beech samples, 17% of spruce samples, and 19% of lysimeter soil water samples. Thus, the seasonal origin of the broadleaf tree water was distinctly out of phase with the precipitation at the time of sampling (27 July to 7 August).

The occurrence of winter precipitation in xylem and soils cannot be simply explained by its carryover in snowpacks or by a lack of summer precipitation. Beech and oak, which used more winter precipitation than spruce, occupied lower elevation (and thus less snowy) sites. Furthermore, for each species, trees in cooler, snowier sites used less winter precipitation (see SOI correlations with temperature and snow fraction in Table 1). Precipitation amounts in Switzerland are distributed relatively evenly throughout the year (Figure S1), with approximately 58% of annual precipitation falling in warmer months (Fig. S1). This was also true of 2015, when the sampling occurred, despite that summer having slightly less precipitation than normal (Fig. S1). The average site received 125 mm (and the driest site received 55 mm) of rain in the 50 days prior to sampling (Fig. S1). Even if the fractional volumetric field capacity were 0.35 (O'Green, 2012) and infiltration occurred as piston flow, such that summer precipitation displaced previously stored waters (both conservative assumptions), 125 mm of summer precipitation would have reached depths of 36 cm (not accounting for evaporation losses). In natural soils, however, infiltrating waters must percolate deeper than they would by piston flow if they mix with previously stored soil moisture or partially bypass the matrix by travelling through macropores (Beven and Germann, 1982; Brooks et al., 2010; Mueller et al., 2014; Sprenger et al., 2016b). Thus, summer precipitation could have reached the depths that contained most of the roots (i.e., 15-40 cm; Fig. 4 and Fig. S2). However, only in the wettest sites did summer precipitation predominantly contribute to tree xylem water (Fig. 2). While winter isotopic signatures have previously been observed in summer xylem of desert plants (Ehleringer et al., 1991), our work demonstrates the widespread use of winter water in mid-summer across diverse humid climates, prompting the question of how much does tree water use depends on summer rainfall.

The study sites a) share similar ecological communities and forest management histories, and b) were sampled by a single field crew over a period of just 12 days, thereby minimizing sampling inconsistencies and facilitating comparisons of





SOI across climatic, topographic and edaphic gradients. Total annual precipitation and other climate metrics associated with water-balance surplus showed statistically significant, positive correlations with the SOI of xylem water and lysimeter soil water (Table S1). Thus, SOI variations in mid-summer xylem water across Switzerland were not only a product of differences in distributions of species with distinct rooting habits. The cross-site trends in lysimeter waters mirrored those of xylem waters (both have stronger winter signatures in drier sites; Fig. 2 and Table 1); this suggests that the xylem water trend does not solely reflect differences in rooting habits, but also trends in mobile soil waters' seasonal origins. Jasechko et al. (2014) similarly observed that aquifers were disproportionately winter-sourced in areas with summer water deficits, probably because under those conditions more summer precipitation evaporates. SOI was also positively correlated with slope and elevation, possibly because these topographic variables co-vary with precipitation (Tables S1 and S2). Surprisingly, correlations between soil characteristics and SOI were weaker and inconsistent (Tables 1 and S1), suggesting that soil texture may be less important than climate in their control over the turnover of plant available water.

Species differed in their water sources, as reflected by differences in their xylem water SOI. Spruce is considered to be shallow-rooted, with its roots mostly occurring in the top 25 cm (Schmid and Kazda, 2002), compared to beech and oak, whose roots are reported to mostly occur in the top 40 cm (Schmid and Kazda, 2002; Thomas and Hartmann, 1998). However, measurements from soil pits excavated in each site show that both maximum and mean rooting depths were broadly similar among beech, spruce, and oak, with mean depths usually ranging between 15 and 35 cm, and maximum depths usually ranging between 50 and 120 cm. (Fig. 4 and Fig. S2). Regardless of these observed similarities in rooting depths, the isotope data show that beech used significantly more winter precipitation than spruce, even within the same plots (mean SOI difference of 0.53, $n$=27 sites, p<0.001 by paired t-test, Fig. 5a; similarly, oak SOI was 0.54 lower than spruce SOI in the one plot where they were paired). In contrast, oak and beech within the same plots used similar waters (mean SOI difference of 0.10, $n$=11 sites, p=0.13 by paired t-test; Fig. 5b). In sites with lysimeters, the soil waters they sampled were significantly less winter-like than beech xylem waters (mean SOI of -0.20 versus -0.90, $n$=16 sites, p<0.001 by paired t-test; Fig. 5c), suggesting that beech roots accessed water sources that were deeper (e.g., saprolite) or more tightly held (e.g., fine pores). These frequently overlooked water sources may be important for storing winter precipitation and supplying summer transpiration (Rempe and Dietrich, 2018). In contrast, lysimeters generally sample more mobile, less tightly held soil water (Brooks et al., 2010; Sprenger et al., 2016). Spruce, unlike beech, appear to use water that is more similar to this mobile water; the SOI of spruce xylem water was statistically indistinguishable from lysimeter soil water at paired sites (mean SOI of -0.27 versus -0.14, $n$=21 sites, p=0.13 by paired t-test; Fig. 5c). Thus, spruce trees used fundamentally different water sources than the two broadleaf species, demonstrating niche partitioning in the rhizosphere across a wide range of soils and climates. Given that the spruce and beech trees had similar rooting depths but used different source waters (Figure 4), we hypothesize that these species' niche separations relate to their relative uptake of water from more vs. less conductive soil pores.

The SOI of xylem water reflects roots' access to water sources with different seasonal dynamics, implying vulnerability to different types of droughts. While deep roots are assumed to mitigate vulnerability to droughts (Ehleringer et



al., 1991; West et al., 2012) by providing access to storages of past precipitation, this will not be the case where deeper substrates lack sufficient storage capacity or are not reliably recharged by infiltrating precipitation (Fan et al., 2017). Here, the data suggest that SOI variations are not solely a reflection of rooting depth differences because rooting depths a) lacked a strong trend across sites (Figure S2), b) were similar among species (Figure 4), and c) were weakly correlated with SOI (Tables 1 and S1). Nonetheless, the xylem water with low SOI directly reflects access to storages that recharge in winter, regardless of whether those waters are deeper or more tightly held. Trees that use stored winter precipitation (e.g., beech and oak in drier regions; Fig. 6) may be less vulnerable to summer precipitation deficits, but more reliant on the soil's capacity to store sufficient amounts of winter water through the growing season. As increasing temperatures result in longer growing seasons (Körner and Basler, 2010), winter water stores may become insufficient to sustain tree growth in regions with seasonal water deficits (Fig. S3). In contrast, mid-summer xylem water with high SOI (e.g., spruce trees in central and southern Switzerland; Fig. 6) directly reflects access to storages with more rapid turnover, in which water from previous seasons has drained away or been displaced by summer precipitation. Spruce's greater use of summer rainfall may explain why it is more sensitive than beech to summer droughts (Brinkmann et al., 2016; Zang et al., 2014). Ultimately, further research is needed to clarify the extent to which seasonal origin signals are attributable to rooting habits versus water-transport processes. Understanding why these spatial and inter-species differences occur and whether they persist are key to understanding their implications for predicting forest vulnerability to droughts.

Beyond these hydrological and ecological insights, our findings have implications for the use of stable isotopes in climate science and ecophysiology, because variations in the seasonal origins of xylem water imply that plant tissue $\delta^2$H or $\delta^{18}$O (frequently used as climatic or ecophysiological proxies) may reflect different seasons in different species, individuals, sites, and years. The $\delta^2$H or $\delta^{18}$O signatures of plant tissue (e.g., cellulose and leaf waxes) reflect the initial $\delta^2$H or $\delta^{18}$O of the source water incorporated into plant tissue, as well as climatically and physiologically controlled fractionation effects (Barbour, 2007). In a variety of isotope applications, it is often useful to attribute the source water to summer rainfall (Lawrence and White, 1984) or mean annual precipitation (Helliker and Richter, 2008); however, we observed waters in trees that had neither a consistent summer signature, nor a consistent mean annual signature (see Fig. S4). Although these xylem waters were sampled at one point in time, they document widespread temporal decoupling between precipitation inputs and plant-water uptake. If, as these results suggest, seasonal origins vary systematically by species and across climatic gradients, accounting for these variations could aid in interpreting plant tissue stable isotopes as environmental proxies.

Variations in SOI also convey information about how soils transport water. Only the wettest sites clearly demonstrated substantial transport of summer precipitation to the rhizosphere. Elsewhere, the summer rainfall apparently did not reach (or potentially bypassed) the relatively shallow depths that contained most of the roots (e.g., 15-40 cm; Figure 3), suggesting that infiltration was not a piston-flow process (e.g., translatory flow), as also previously argued by Brooks et al. (2010). Further evidence for the lack of translatory flow can be observed in the scarcity of soil waters (sampled by lysimeters or taken up by roots) with a strong summer signature (Figure 2); this indicates that summer precipitation must either mix with or bypass the





storage. Our findings differ from the ecohydrologic separation in a Mediterranean climate described by Brooks et al. (2010), in which infiltrating water refilled pore spaces when soils were dry and otherwise bypassed the rhizosphere when soils were wet. In contrast, our results indicate that recent precipitation refilled rhizosphere pore spaces more in wetter sites than in drier sites. The low SOI of xylem water in beech and oak trees implies that little summer precipitation traveled to their roots, likely because it either bypassed the soil matrix or was retained in near-surface soils before quickly returning to the atmosphere (e.g., by evaporation and understory transpiration). A recent modelling study (Brinkmann et al., 2018), that did not account for evaporation effects, computed that roughly 50% of tree water use came from summer precipitation at one site in Switzerland. In contrast, our measurements from over 900 trees across a network of diverse sites empirically show that the majority of mid-summer tree xylem, and by extension rhizosphere soils, contain only small contributions from summer precipitation. These measurements imply that the turnover of water (and thus flushing of solutes) in these trees' rooting zones must be remarkably small in summer.

Although SOI values do not precisely record water age, the widespread presence of winter precipitation in summer soils indicates that these waters often resided in soils for months with minimal mixing, suggesting that summer precipitation flows preferentially through those soils. We can explore these flow processes through a back-of-the-envelope calculation. Mean transit time can be calculated hydrometrically as storage divided by flux (also often referred to as mean turnover time). We conservatively estimate the storage above the rooting zone to be 10.5 cm of water (most roots are above 30 cm, shown in Figure 3, multiplied by the maximum field capacity of 0.35). We estimate the mean flux to the roots to be 1.36 mm per day, calculated as precipitation minus evaporation using the precipitation (2.51 mm per day) and mean PET (4.6 mm per day) across the sites in the 50 days prior to sampling, and with evaporation assumed to be 0.25 of PET, which is conservative as an estimated fraction of evaporation over actual evapotranspiration in full-canopy forest (Schlesinger and Jasechko, 2014). This yields an estimated hydrometric mean transit time (or turnover time) of 77 days in summer (and it must be substantially shorter in spring or winter because PET is lower and precipitation rates are similar). If mean transit times are 77 days and soil water storages are composed of waters with undiluted, mid-winter precipitation isotope values, then stored waters must be substantially older than the mean transit time. This contrast between storage ages and the mean transit time would suggest that soil water flows are neither well-mixed, nor translatory and instead are preferential (*sensu* Berghuijs and Kirchner, 2017), although it remains unclear how fully roots sample and reflect the age of soil water storage. Regardless, the empirical insights shown here – specifically, trends in the seasonal origins of water in soils across climates, and differences in the use of recent precipitation versus older stored precipitation among species – may find application in better parameterizations of plant uptake of water from dynamic storages in gridded hydrological or ecological models.

More broadly, the analytical framework introduced here provides a new tool for applying stable isotope data to explore a wide variety of ecological and hydrological processes. Here, the seasonal origin analysis aids in describing plant-soil-water interactions and how they vary across landscapes; specifically, examining the seasonal origin of tree-water revealed (1) the consistent inter-species differences in rhizosphere water niches, (2) the long residence times of root-zone soil moisture





in summer, and (3) the need to consider the over-expression of different seasons' precipitation when interpreting plant-tissue isotopes. We suspect that insights may also be revealed through applying this seasonal origin index analysis to groundwater (*sensu* Jasechko et al., 2014), stream water, or even plant and animal tissues.

## 4. Summary and Conclusions

The seasonal origins of precipitation used by trees, which reflect the interplay between infiltration dynamics and root distributions, have not previously been systematically investigated. We used a spatially extensive snapshot sample of xylem water from Swiss forest plots to quantify the seasonal origins of water used by trees in midsummer. Xylem waters in 918 trees from 182 sites (and soil lysimeter water from a subset of these sites) were sampled and analyzed for $\delta^{18}O$ and $\delta^2H$ by a single team using consistent methods (thereby avoiding many uncertainties that are common to meta-analyses). By applying a new index that characterizes the occurrence of summer versus winter precipitation in these xylem samples, we show that trees mostly used winter precipitation in mid-summer in all but the wettest regions of Switzerland (Figure 1). Summer precipitation isotope signatures were uncommon in shallow soils, deep soils, and tree xylem (Figure 2), suggesting that infiltrating precipitation does not simply displace stored soil waters. There was consistent partitioning in the water sources used by different species (Figures 3): beech and oak almost entirely used winter precipitation, whereas spruce used more mixed sources that were isotopically similar to the water extracted by suction lysimeters (presumably from more conductive pores). The widespread prevalence of winter precipitation in mid-summer tree xylem suggests that (a) the turnover of water (and thus flushing of solutes) in these trees' rooting zones must be remarkably small in summer and (b) plant-tissue isotope proxies may not consistently capture summer climate signals. These findings conflict with common assumptions on tree water use and provide empirical support for developing more realistic representations of root-soil-water interactions.

**Acknowledgments**: We thank N. Engbersen, C. Romero, L. Schmid, and the Institute for Applied Plant Biology team for assistance with sample collection and processing, Wouter Berghuijs and Julia Knapp for comments on the manuscript, and Wouter Berghuijs and Paolo Benettin for useful discussion during the writing process. The forest departments of the cantons AG, BE, BL, BS, GR, SO, TG, ZH and ZG, as well as the environmental offices of Central Switzerland, funded the tree sampling. This project was funded by a Swiss Federal Office of the Environment agreement with G.R. Goldsmith and J.W. Kirchner. G.R. Goldsmith was supported by funding from the European Commission's Seventh Framework Program (FP7/2007-2013) under grant agreement number 290605 (COFUND: PSI-FELLOW) while at the Paul Scherrer Institute.

**Author contributions:** SA conceived and executed the analysis, with input from GG and JK. SA, JK and GG wrote the paper. SB, GG, and RS initiated the project and coordinated the 2015 field and lab work. SB leads the long-term measurement network.



**Competing Interests:** The authors declare that they have no conflict of interest.

**Data and materials availability:** The data that support the findings of this study are currently available from the corresponding author upon reasonable request, and will be made available on Dryad (https://doi.org/xxxxxx) upon acceptance. We acknowledge data contributions by International Atomic Energy Association and GNIP contributors as well as Swiss, German, and Austrian federal monitoring agencies (Swiss Federal Office of the Environment (FOEN) and its NAQUA program, MeteoSwiss, Austrian Network of Isotopes in Precipitation, Austrian Zentralanstalt für Meteorologie und Geodynamik, and Deutscher Wetterdienst).

**Supplementary Materials**

Table S1 Correlations between site characteristics and xylem water and lysimeter soil water SOI.

Table S2 Multiple regression models of xylem water and lysimeter soil water SOI.

Figure S1 Precipitation amount at the study sites.

Figure S2 Soil, lysimeter, and root depths across the sites.

Figure S3 Mapped distributions of selected site characteristics for the 182 study sites.

Figure S4 Distributions of measured $\delta^2$H of tree xylem water across individuals, relative to the long-term, site-specific, volume-weighted, mean isotopic composition of precipitation (evaluated for 2007-2015).

Figure S5 Effects of compensating for evaporative enrichment.

Figure S6 Pairwise comparisons of non-fractionation-compensated $\delta^2$H for sites where a) spruce and beech are collocated, b), oak and beech are collocated, and c), trees and lysimeters are collocated.

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





**Table 1 Spearman (rank) correlations between site characteristics and xylem water and lysimeter soil water SOI, where statistically significant correlations are indicated with bold fonts (p < 0.05) and gray shading (p < 0.01); see Table S1 for a more extensive correlation table.**

| Site characteristics | Beech SOI | Spruce SOI | Oak SOI | Shallow soil SOI | Deep soil SOI |
|---|---|---|---|---|---|
| Soil depth | -0.11 | -0.21 | 0.08 | **-0.52** | -0.23 |
| Stone fraction in top 50 cm | **0.34** | **0.26** | 0.06 | 0.34 | 0.01 |
| Clay fraction in top 50 cm | 0.15 | -0.02 | 0.02 | -0.20 | 0.10 |
| Mean root depth | 0.14 | 0.01 | 0.28 | -0.17 | **-0.61** |
| Site elevation | **0.42** | **0.57** | -0.01 | **0.41** | 0.26 |
| Mean temperature | -0.12 | **-0.48** | 0.19 | **-0.42** | -0.29 |
| Snow fraction | 0.12 | **0.42** | -0.24 | **0.49** | 0.32 |
| Previous 50-day's precip. | **0.22** | **0.25** | -0.25 | 0.28 | 0.05 |
| Annual precipitation | **0.50** | **0.51** | 0.08 | 0.37 | 0.20 |
| Summer 2015 precip.− PET | **0.43** | **0.59** | 0.09 | **0.43** | 0.22 |





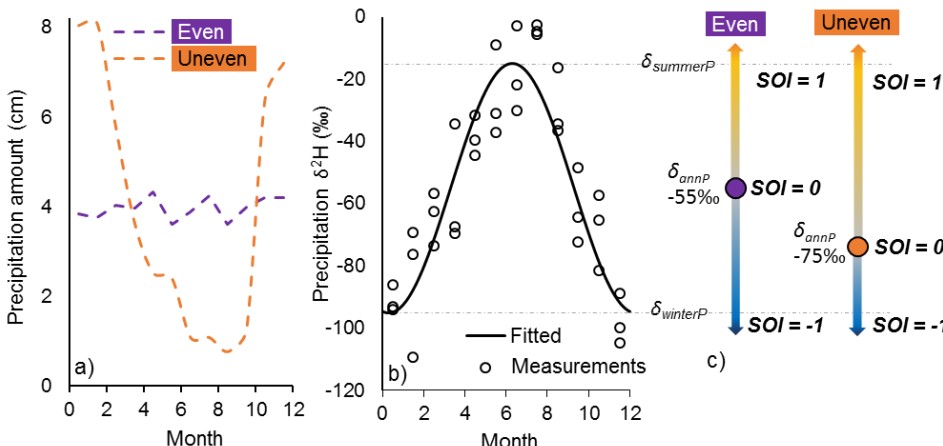

**Figure 1 Calculation of the seasonal origin index (SOI). As a hypothetical example, consider one site that receives equal precipitation amounts throughout the year, and another site that receives more precipitation in winter (a), but both have the same seasonal isotopic cycle (b). In this example, the volume-weighted average precipitation is -75 ‰ $\delta^2$H in the uneven-precipitation site, and -55 ‰ $\delta^2$H in the even-precipitation site (c); these values mark SOI=0. Thus, if water with -60 ‰ $\delta^2$H was observed in xylem in the uneven-precipitation site, SOI would be positive, indicating that each mm of rain that fell during the summer made a larger contribution to xylem water than each mm of rain that fell during the winter (even though, owing to the greater precipitation in winter, winter precipitation made up more than 50% of the xylem water). Panel b also shows how the seasonal precipitation isotope cycle is defined by a fitted sinusoid, such that the amplitude captures typical summer and winter peaks, and not the absolute bounds of possible values (i.e., SOI of soil or xylem water can be higher than 1.0 or lower than -1.0).**



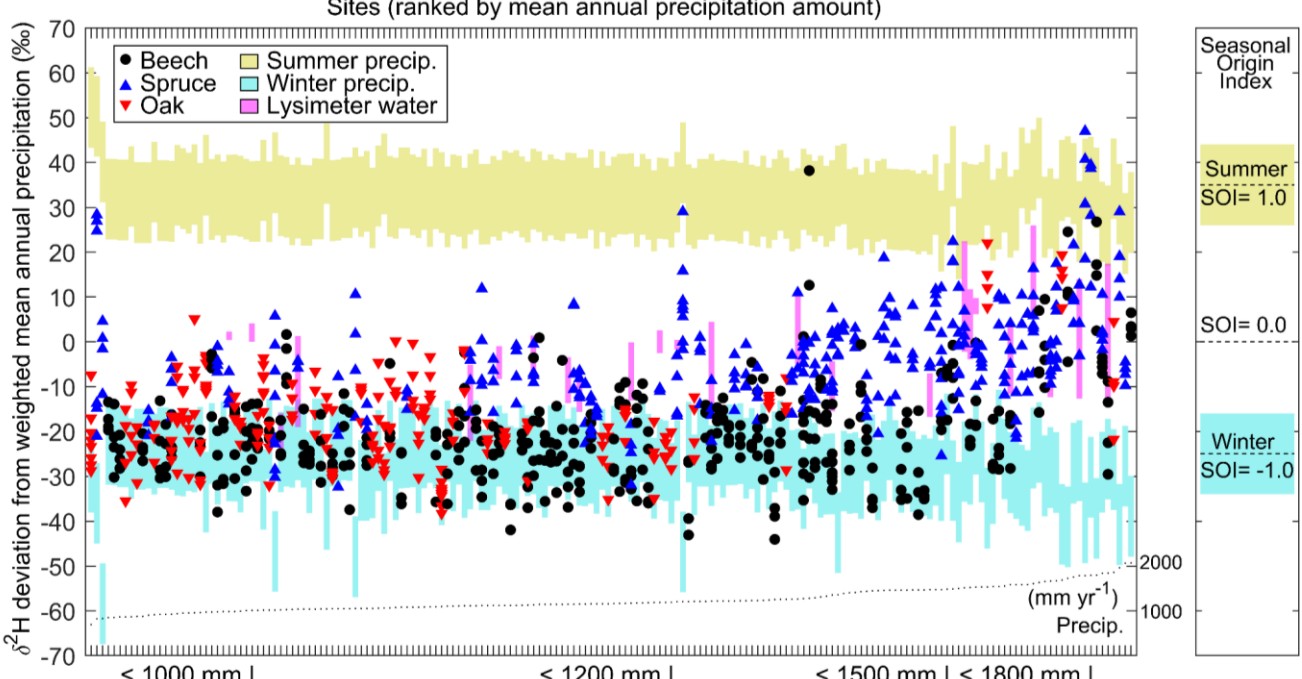

**Figure 2 Isotope ratios of xylem water and lysimeter soil water, compared to site-specific seasonal isotope cycles in precipitation.**
**Fractionation-compensated isotope ratios for xylem and soil lysimeter water are plotted as deviations from each site's volume-weighted annual precipitation δ²H for the two years prior to the summer 2015 sampling. Typical isotope values for summer and winter precipitation are shaded (see Methods). Magenta bars show the δ²H range of lysimeter soil water in depth profiles. Sites are ranked by mean annual precipitation amount (see dotted black line and labels below the horizontal axis). The panel on the right depicts how isotope values translate to Seasonal Origin Index values. Trees in all but the wettest sites mostly use water that isotopically resembles winter precipitation (i.e., negative SOI).**



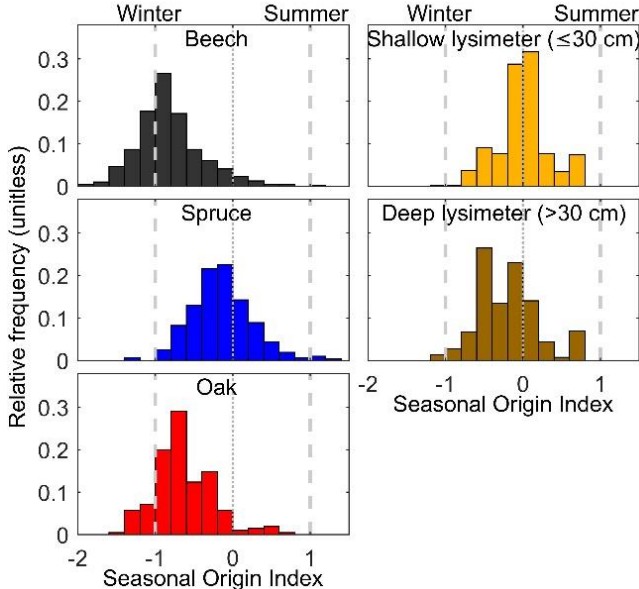

**Figure 3 Distributions of the seasonal origin of water in soils and trees across Switzerland in mid-summer. Beech and oak xylem show a predominance of winter precipitation. Soil porewaters (sampled by suction lysimeters) and spruce xylem indicate mixtures of precipitation from multiple seasons. Seasonal Origin Index values below 1 reflect waters that are more isotopically negative than typical winter precipitation (estimated by sinusoidal fitting of precipitation patterns; see Materials and Methods).**





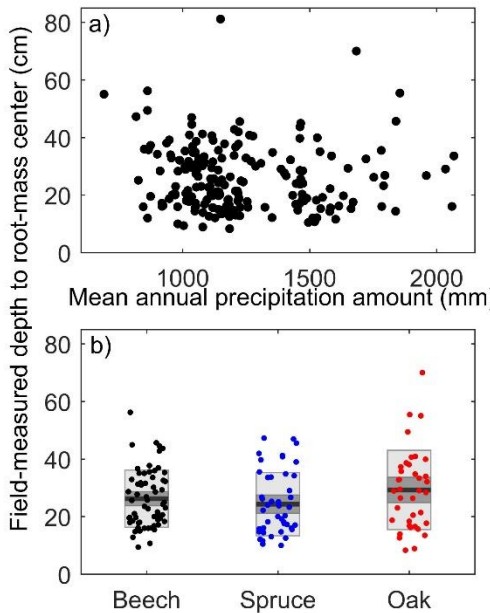

**Figure 4 Field measured root depths (a) versus site mean annual precipitation amount and (b) by species for single-species plots. Depth to root-mass center is the mean rooting depth weighted by density of roots per horizon. (a) Its variations were not linearly related to mean annual precipitation ($R^2 < 0.01$, $p = 0.40$). (b) The box plots show means (black line) ± one standard error (medium gray), and ± one standard deviation (light gray), of maximum (c) and mean (e) root depths, for the stands that only contained one of the study species; the data suggest that the three species have similar average rooting depths across the sampled sites. See Figure S2 for more details figures on soil and root depths.**





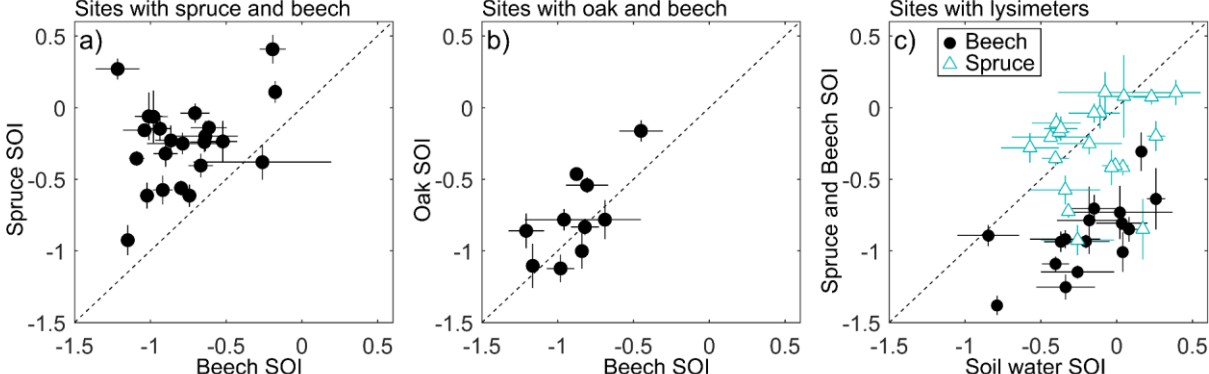

**Figure 5 Pairwise comparisons of Seasonal Origin Index values for sites where a) spruce and beech are collocated, b), oak and beech are collocated, and c), trees and lysimeters are collocated. 1:1 lines are plotted for reference, highlighting that (a) spruce used more summer-sourced water than beech, whereas (b) beech and oak used similar water supplies. Additionally, (c) spruce used water similar to lysimeter soil water, unlike beech. Symbols indicate site means with error bars representing one standard error of the mean, attributable to intra-site variability.**




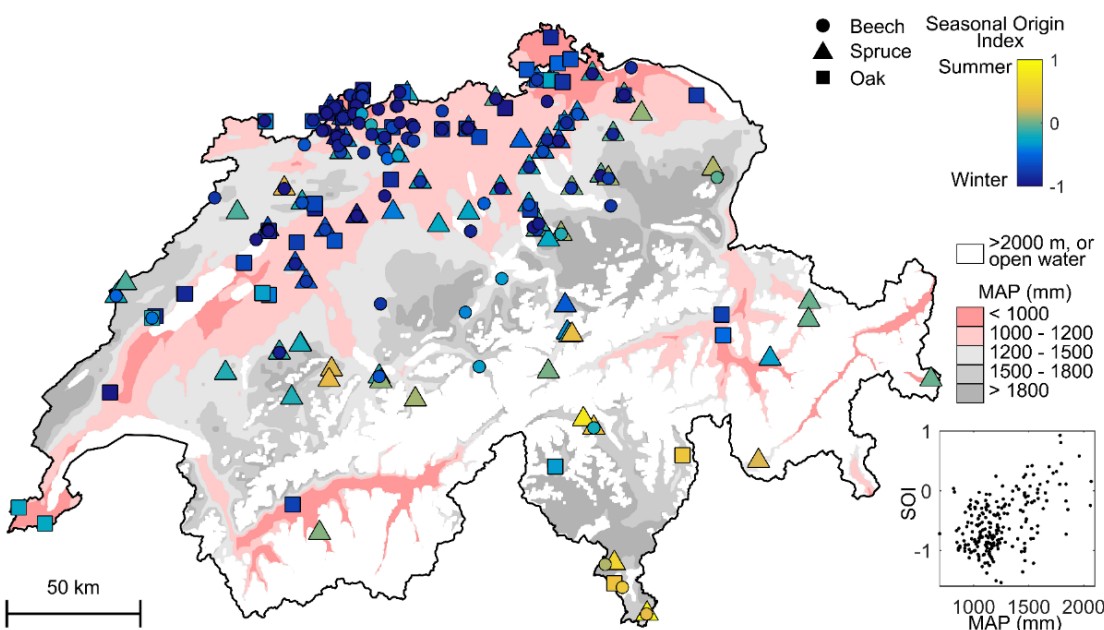

**15    Figure 6 Variation in seasonal origins of tree xylem water with mean annual precipitation (MAP) across Switzerland. Open water and elevations > 2000 m a.s.l. are excluded. In all but the wettest regions, the Seasonal Origin Index shows a predominance of winter precipitation in tree xylem.**