# Peer review of "Seasonal origins of soil water used by trees"

_Hydrology and Earth System Sciences, 2018_

## Referee Comment (RC1) · Anonymous Referee #1 · 5 Dec 2018

This manuscript describes a seasonal index developed using stable water isotope data from precipitation to characterize the seasonal water source origin (i.e., winter or summer) of soil water or tree xylem water. The data represent almost 200 forest sites in Switzerland, which is an impressive sample of systems. Using the seasonal origin index, the authors describe that winter precipitation was the predominant water source that was stored in soil water or that trees were using during the mid-summer. This manuscript contributes to the ongoing discussion in the ecohydrological community about how plants extract water from the soil storage and the so called ecohydrologic separation between plant water and water that contributes to streamflow or groundwater recharge. In my opinion, this is an important paper and will generate discussion. The manuscript is very well written and the analysis is pretty thorough. I also appreciate the detailed information that is provided by the supplemental information. I would recommend publication of this work. I have a few comments, mostly editorial, that I

hope will help improve the manuscript.

One minor issue is that the authors consider that suction lysimeters sample mobile water. In a relative sense compared to other sampling methods this is largely true. However, mobility is a continuum and it might be better to relax that dichotomy. For one, the suction that is used is rather high compared to other studies that are attempting to sample "mobile" water (e.g., -10 to -30kPa, especially considering that many use the somewhat arbitrary definition of field capacity as ∼-30 kPa). Also, it should be noted that even though the suction was applied for a month or so, most of the water was likely sampled within the first day or so and that the longer suction is maintained, the more the sample water represents the applied tension (i.e., finer pore space; see Severson & Grigal, Everett & McMillion, and Weihermuller et al. for example).

The analysis with rooting distribution is interesting and it was certainly an achievement to collect observations from so many sites. The method, however, is likely biased by coarse roots and may not reflect the finer roots that participate in water uptake. Some discussion of this is encouraged. It also wasn't clear what depths were involved in the root survey, i.e., only that it was by horizon. I understand that horizon depths vary by site, but please provide more information, for example, were O horizons surveyed? In general, there are more fine roots and root distribution is highest in the upper 10 cm of soil, which would typically include the O horizon.

Line comments:

P2, Line 6, comma after thus

P2, Line 14: after water ages, perhaps cite a study where storage selection functions are used to describe how ET is selected from storage and hence represents water of varying age. The Sprenger et al. citation in the sentence above is close by there might be something else that work better (e.g., Botter et al. 2011 show this idea conceptually with eqn. 6, but maybe there is a field study showing something similar). Also, it's not just fast/slow pathways, but as is discussed later, it is dependent upon rooting

distribution.

P2, Line 23: more mobile relative to what?

P3, Line 8: In addition to texture, some additional characteristics related to soil (e.g., all same soil order or type?) and parent material would be helpful.

P4, Line 10: Please give range of depths here.

P6, Line 22: Perhaps note parenthetically that Pearson is only in the supplemental information.

P7: Line 9: Could snow canopy interception make the difference here?

P7, Lines 21-22: Surprising that most roots are not nearer the surface. Jackson et al. suggest that the top 10 cm contain about 25% of the roots. Could this be the bias of survey more coarse roots than fine?

P7, Line 33: There are about 8-9 places in the manuscript where using the possessive seems awkward and the sentence could be recast. Please consider reducing the use of the possessive.

P8, Line 4: There are other soil properties that could be quite relevant, e.g., structure.

P8, Line: Be specific and refer to suction cup lysimeters and perhaps give a few sampling methods that do in fact sample water that is less mobile.

P8, Line 24-25: Could this be related to litter interception since spruce litter is quite different than the broad-leaved species? Also, here and the next sentence are two more examples where the sentences could be slightly recast to remove the use of a possessive.

References:

Botter, G., Bertuzzo, E., & Rinaldo, A. (2011). Catchment residence and travel time distributions: The master equation. Geophysical Research Letters, 38(11).

[Figure]

Everett, L. G., & McMillion, L. G. (1985). Operational ranges for suction lysimeters. Groundwater Monitoring & Remediation, 5(3), 51-60.

Jackson, R.B., J. Canadell, J.R. Ehleringer, H.A. Mooney, O.E. Sala, and E.D. Schulze. 1996. A global analysis of root distributions for terrestrial biomes. Oecologia 108:389–411.

Severson, R. C., & Grigal, D. F. (1976). Soil solution concentrations: effect of extraction time using porous ceramic cups under constant tension. JAWRA Journal of the American Water Resources Association, 12(6), 1161-1170.

Weihermüller, L., Kasteel, R., Vanderborght, J., Pütz, T., & Vereecken, H. (2005). Soil water extraction with a suction cup. Vadose Zone Journal, 4(4), 899-907.

---

## Referee Comment (RC2) · Anonymous Referee #2 · 7 Jan 2019

The manuscript "Seasonal origins of soil water used by trees" by Allen et al. presents a study from Switzerland showing that in summer beech and oak trees preferably used soil water that was recharged by winter precipitation in contrast to spruce trees that used soil water with more contributions from summer precipitation. The topic is highly relevant because it gives details about the importance of winter and summer precipitation for the growth of three tree species. The manuscript is very well written and most of the conclusions are supported by the data. Below are given some comments relevant for the interpretation of the soil water data and for the conclusions on water uptake by the trees.

Abstract, page 1, line 27/28: there are very good concepts of how water flows through soils, and this statement is not a consequence of the findings of this manuscript because soil water fluxes were not investigated. Therefore, I suggest deleting this statement

Introduction, page 2, line 20: the general literature introduction is rather short. There are studies from humid regions, and I suggest already mentioning some and their findings in the introduction and not only in the discussion (e.g. Brinkmann et al (cited in the study) or Dubbert et al., accepted (doi: 10.1111/nph.15670)).

Page 4, lines 6-15: The applied tension of 60-70 kPa results in sampling water from a large range of different pores. This includes both water from larger and smaller pores, and thus more and less mobile water. In some of the very sandy soils, this could even be the entire pore water pool (depending on the water retention function of these soils). Therefore, it is rather speculative which pore regions the sampled water is representative for. This should be considered in the entire interpretation of the data.

Page 8, lines 3-5: to be more precise, I would add "for humid regions" here.

Page 8, line 18: there are no supporting data that water from finer pores had lower isotopic ratios compared to the pore water extracted with the lysimeters.

Page 8, line 19: see previous comment on soil water sampling

Page 9, line 1: I agree that these trees may be less vulnerable to summer precipitation deficits; however, only when considering short-term effects, as outlined in the next lines.

Page 9, lines 30-32: already mentioned at the beginning of this paragraph (lines 22-28) and thus redundant

Page 10, line 6: considering general low flow velocities in soils, I don't think that it is very surprising to find winter precipitation in these soils.

Page 10, line 11-19: 1) what is the average measured water content in these soils, in summer and in winter? The assumption of a field capacity of 0.35 for silty soils could be okay but is too larger for sandy soils. 2) the 77 days only refers to the time when all water (summer precipitation) is taken up by the roots. The total soil water balance (including up-/downward fluxes below the root zone) need to be considered for an

estimation of mean transit times. 3) mean water contents are higher in winter/spring compared to summer strongly influencing the transit time of water. The back-of-the envelope calculation needs further support by measured data and calculation of water fluxes and could be part of a more thorough, next investigation. From the presented data a more detailed interpretation of soil water fluxes is not possible and cannot be generalized over all investigated sites considering the wide range of investigated soils (see variation of soil texture).

page 11, line 12: see earlier comment on soil water sampling

––––––––––––––––––––––––

---

## Author Comment (AC1) · 9 Jan 2019

This manuscript describes a seasonal index developed using stable water isotope data from precipitation to characterize the seasonal water source origin (i.e., winter or summer) of soil water or tree xylem water. The data represent almost 200 forest sites in Switzerland, which is an impressive sample of systems. Using the seasonal origin index, the authors describe that winter precipitation was the predominant water source that was stored in soil water or that trees were using during the mid-summer. This manuscript contributes to the ongoing discussion in the ecohydrological community about how plants extract water from the soil storage and the so called ecohydrologic separation between plant water and water that contributes to streamflow or groundwater recharge. In my opinion, this is an important paper and will generate discussion.

The manuscript is very well written and the analysis is pretty thorough. I also appreciate the detailed information that is provided by the supplemental information. I would recommend publication of this work. I have a few comments, mostly editorial, that I hope will help improve the manuscript.

**Thank you for your kind remarks and your editorial comments.**

One minor issue is that the authors consider that suction lysimeters sample mobile water. In a relative sense compared to other sampling methods this is largely true. However, mobility is a continuum and it might be better to relax that dichotomy. For one, the suction that is used is rather high compared to other studies that are attempting to sample "mobile" water (e.g., -10 to -30kPa, especially considering that many use the somewhat arbitrary definition of field capacity as ∼-30 kPa). Also, it should be noted that even though the suction was applied for a month or so, most of the water was likely sampled within the first day or so and that the longer suction is maintained, the more the sample water represents the applied tension (i.e., finer pore space; see Severson & Grigal, Everett & McMillion, and Weihermuller et al. for example).

**We agree that mobility is a continuum. The conceptual discretization of this continuum into more and less mobile fractions can facilitate communication, although there is some risk of over-simplifying. We have followed the reviewer's advice, and relaxed the binary language. On page 4, we change "To determine the $\delta^{18}O$ and $\delta^{2}H$ ratios of mobile soil waters, samples were collected from suction lysimeters" to "We also determine the $\delta^{18}O$ and $\delta^{2}H$ ratios of soil waters accessed by suction lysimeters, which tend to sample the more mobile soil waters (i.e., in contrast to water under high tension or in tight pore spaces)". In the other instance where we referred to "the mobile fractions", we change it to "the more mobile fraction" (page 4 and 8) so that we do not imply that there is a strict partitioning. In the revised manuscript, we also elaborate upon the likely timing implications of the lysimeter sampling (and now add citations).**

The analysis with rooting distribution is interesting and it was certainly an achievement to collect observations from so many sites. The method, however, is likely biased by coarse roots and may not reflect the finer roots that participate in water uptake. Some discussion of this is encouraged. It also wasn't clear what depths were involved in the root survey, i.e., only that it was by horizon. I understand that horizon depths vary by site, but please provide more information, for example, were O horizons surveyed? In general, there are more fine roots and root distribution is highest in the upper 10 cm of soil, which would typically include the O horizon.

**The root density metric that we report is actually fine root density, which we have now clarified. We also now comment on the possibility of missing roots on page 11, and we now mention that not all roots transport water equally. The depth information is provided in Figure S2, which we now refer to in the Methods section.**

**For the reviewer: approximately 1/3 of the soils were recorded to have an O horizon, which generally occupied the top 1-10 cm (median of 5, among sites with O horizons) and had fine root densities that were generally classified under categories "strong", "very strong" or "extremely strong". These data will be available in the archived data files.**

Line comments:

P2, Line 6, comma after thus

**A comma has been added.**

P2, Line 14: after water ages, perhaps cite a study where storage selection functions are used to describe how ET is selected from storage and hence represents water of varying age. The Sprenger et al. citation in the sentence above is close by there might be something else that work better (e.g., Botter et al. 2011 show this idea conceptually with eqn. 6, but maybe there is a field study showing something similar). Also, it's not just fast/slow pathways, but as is discussed later, it is dependent upon rooting distribution.

**Botter et al's equation 6 is relevant, and we suggest that our findings here could be considered with respect to age conservation concepts. We now cite it as recommended.**

P2, Line 23: more mobile relative to what?

**By adding the word "fraction", we believe it is now clear that "more" is in comparison to the other fractions of water held in a given soil.**

P3, Line 8: In addition to texture, some additional characteristics related to soil (e.g., all same soil order or type?) and parent material would be helpful.

**While this is a fair question, it goes beyond the scope of our study because there are so many different soil types and parent materials across our domain. It would require considering these sites on a case-by-case basis, which is not our objective. For example, some are calcareous, some are highly clayey, not all have B horizons, and not all have O horizons. Geologic substrates include moraine deposits, river gravels, marl, loess, clay deposits, sandstone, dolomite, gneiss, and limestone. These soils span numerous soil orders and parent materials. Last, there are also reporting challenges that arise from subjectivity and language-specificity of soil classifications.**

**Because information related to soil type may be of interest to those interested in reusing our isotope data, the data file will also include taxonomic characterizations of the horizons (from which soil types can be inferred) and by-horizon data on soil textures and root densities. Coordinates for these sites are included in the open-access data, and they can be linked with publicly available geologic data from SwissTopo (Swiss Federal Office of Topography, Wabern, Switzerland).**

P4, Line 10: Please give range of depths here.

**I think that the reviewer is referring to the following paragraph (line 15), and requesting the depths of the lysimeters. We have elaborated and added a summary description of the depths. We also refer readers to data file that will show the lysimeter depths for each site.**

P6, Line 22: Perhaps note parenthetically that Pearson is only in the supplemental information.

**We have added a note that the Pearson coefficients are in the supplemental information.**

P7: Line 9: Could snow canopy interception make the difference here?

**It could certainly have an effect, but we are not aware of any studies showing snow interception effects that could have this large of an influence. Thus we prefer to not speculate here. No changes have been made.**

P7, Lines 21-22: Surprising that most roots are not nearer the surface. Jackson et al. suggest that the top 10 cm contain about 25% of the roots. Could this be the bias of survey more coarse roots than fine?

**Maybe our original text was not clear enough; our results are actually consistent with those of Jackson et al. The quoted depth of 15-40 cm is not the depth that contains most of the roots, but the threshold above which most roots are located. We have revised our language to clarify that "Most roots occur above 15-40 cm, depending on the site".**

P7, Line 33: There are about 8-9 places in the manuscript where using the possessive seems awkward and the sentence could be recast. Please consider reducing the use of the possessive.

**We have removed about half of them, as recommended. Some have been retained in the interests of clarity.**

P8, Line 4: There are other soil properties that could be quite relevant, e.g., structure.

**We agree, but unfortunately no structural traits have been characterized in these soils. No changes have been made.**

P8, Line: Be specific and refer to suction cup lysimeters and perhaps give a few sampling methods that do in fact sample water that is less mobile.

**Now in several places throughout the manuscript, we remind readers that "lysimeter soil water" refers to waters sampled by *suction* lysimeters.**

P8, Line 24-25: Could this be related to litter interception since spruce litter is quite different than the broad-leaved species? Also, here and the next sentence are two more examples where the sentences could be slightly recast to remove the use of a possessive.

**In this paragraph, we are referring to mixed forest sites, where their root distributions likely overlap (at least in a lateral domain). However, it is likely that differences in the evaporation of water prior to its infiltration to the depths of roots could cause inter-site variations that we observed (as described in the following paragraph). We now mention interception in the next paragraph to cue the readers towards considering that interception and soil water evaporation occur differently.**

References:

Botter, G., Bertuzzo, E., & Rinaldo, A. (2011). Catchment residence and travel time distributions: The master equation. Geophysical Research Letters, 38(11).

Everett, L. G., & McMillion, L. G. (1985). Operational ranges for suction lysimeters. Groundwater Monitoring & Remediation, 5(3), 51-60.

Jackson, R.B., J. Canadell, J.R. Ehleringer, H.A. Mooney, O.E. Sala, and E.D. Schulze. 1996. A global analysis of root distributions for terrestrial biomes. Oecologia 108:389–411.

Severson, R. C., & Grigal, D. F. (1976). Soil solution concentrations: effect of extraction time using porous ceramic cups under constant tension. JAWRA Journal of the American Water Resources Association, 12(6), 1161-1170.

Weihermüller, L., Kasteel, R., Vanderborght, J., Pütz, T., & Vereecken, H. (2005). Soil water extraction with a suction cup. Vadose Zone Journal, 4(4), 899-907.

---

## Author Comment (AC2) · 9 Jan 2019

**Response to Anonymous Referee #2**

We thank Referee #2 for these helpful comments (listed below in standard font). Our responses and planned changes to the manuscript are listed below in **bold font**.

*Anonymous Referee #2*

The manuscript "Seasonal origins of soil water used by trees" by Allen et al. presents a study from Switzerland showing that in summer beech and oak trees preferably used soil water that was recharged by winter precipitation in contrast to spruce trees that used soil water with more contributions from summer precipitation. The topic is highly relevant because it gives details about the importance of winter and summer precipitation for the growth of three tree species. The manuscript is very well written and most of the conclusions are supported by the data. Below are given some comments relevant for the interpretation of the soil water data and for the conclusions on water uptake by the trees.

Abstract, page 1, line 27/28: there are very good concepts of how water flows through soils, and this statement is not a consequence of the findings of this manuscript because soil water fluxes were not investigated. Therefore, I suggest deleting this statement

**The presence of different seasons' precipitation in soils and tree water uptake directly reflect timescales of transport, mixing, and storage in soils. Specifically, the last three paragraphs of Section 3 show how our isotope measurements can be used to draw inferences about soil water fluxes. In any case, the statement in question was simplified and now reads "These results challenge common assumptions concerning how water flows through soils and is accessed by trees."**

Introduction, page 2, line 20: the general literature introduction is rather short. There are studies from humid regions, and I suggest already mentioning some and their findings in the introduction and not only in the discussion (e.g. Brinkmann et al (cited in the study) or Dubbert et al., accepted (doi: 10.1111/nph.15670).

**There are hundreds of studies that discuss water uptake, and the ones we have selected are not arbitrary. Here we are interested in temporal (rather than spatial) separation among water sources to plants, for which the relevant literature is indeed small. We have now mentioned Brinkmann et al. in the second paragraph of the introduction. We have also added a citation of Dubbert et al., although we point out that this paper only became available online two months after our paper was submitted, and four days before the review was posted.**

Page 4, lines 6-15: The applied tension of 60-70 kPa results in sampling water from a large range of different pores. This includes both water from larger and smaller pores, and thus more and less mobile water. In some of the very sandy soils, this could even be the entire pore water pool (depending on the water retention function of these soils). Therefore, it is rather speculative which pore regions the sampled water is representative for. This should be considered in the entire interpretation of the data.

**We have certainly considered that there are uncertainties concerning what water is extracted from soils. Note that we do not refer to soil water, but instead "lysimeter soil water" (although we remind the reviewer that trees are also samplers of soil water). We also remind the reviewer that with few exceptions, we use these lysimeter water values as ancillary data to bolster the arguments made using the xylem water samples. In our experience, hydrologists have widely diverging opinions on what information is conveyed by suction lysimeter samples; for example, the comments by Reviewer 1 and Reviewer 2 contrast on this point. We believe our assumptions to be very conservative, compared to those by others, and we recognize that some researchers**

**may differ in opinion. Nonetheless, we have expanded discussion of lysimeters in the methods section to comment on this debate and clarify our assumptions.**

**Although the *tension* applied by a lysimeter will be felt in pores of all sizes, but how efficiently those pores are *sampled* will depend on their conductivity and connectivity. Large pores conduct water much more readily (Hagen–Poiseuille equation), and we do not agree that it is speculative to assume that the smallest pore spaces will generally contribute less to the lysimeter water. We now refer to the "the more mobile fraction" accessed by lysimeters, because it is by definition true (i.e., the water that moved to the lysimeter is certainly more mobile what was left behind and not sampled). We agree that there are no absolute pore size thresholds, which we now state. These concepts have been discussed elsewhere (now cited in the paper), and they are not the focus of this study. Regardless, none of our conclusions depend on subtle interpretations of suction lysimeter data.**

Page 8, lines 3-5: to be more precise, I would add "for humid regions" here.

**The statements in question concern our specific sites in Switzerland, so adding "for humid regions" would imply a global extrapolation that we do not make here. No changes are made.**

Page 8, line 18: there are no supporting data that water from finer pores had lower isotopic ratios compared to the pore water extracted with the lysimeters.

**We don't understand the comment. There is no statement about lysimeter data or pore sizes on line 18.**

Page 8, line 19: see previous comment on soil water sampling

**We don't understand the comment. Line 19 does not concern soil water sampling.**

Page 9, line 1: I agree that these trees may be less vulnerable to summer precipitation deficits; however, only when considering short-term effects, as outlined in the next lines.

**We agree (which is why the next lines are written the way that they are). No changes are made.**

Page 9, lines 30-32: already mentioned at the beginning of this paragraph (lines 22-28) and thus redundant

**In the online pdf, 30-32 and 22-28 are different paragraphs, so we do not understand the specific comment.**

Page 10, line 6: considering general low flow velocities in soils, I don't think that it is very surprising to find winter precipitation in these soils

**The point is not that the soils contain some winter precipitation, but that tree uptake is often *dominated* by winter precipitation. We have shown using simple mass balance models that translatory flow assumptions are inconsistent with the persistence of winter precipitation in the rooting zones of these soils. No changes are made.**

Page 10, line 11-19: 1) what is the average measured water content in these soils, in summer and in winter? The assumption of a field capacity of 0.35 for silty soils could be okay but is too larger for sandy soils. 2) the 77 days only refers to the time when all water (summer precipitation) is taken up by the roots. The total soil water balance (including up-/downward fluxes below the root zone) need to be considered for an estimation of mean transit times. 3) mean water contents are higher in winter/spring compared to summer strongly influencing the transit time of water. The back-of-the envelope calculation needs further support by measured data and calculation of water fluxes and could be part of a more thorough, next investigation. From the presented data a more detailed interpretation

of soil water fluxes is not possible and cannot be generalized over all investigated sites considering the wide range of investigated soils (see variation of soil texture).

**We intentionally call this a back-of-the-envelope calculation. It is not a fully elaborated simulation model and is not intended as such. The model that the reviewer is suggesting would require even more assumptions (e.g., regarding how spatially representative soil moisture measurements are, how well texture defines the storage capacity, and so on). While such a model could be useful, it would still be uncertain, and its assumptions would be much less transparent. Thus, we believe that this back-of-the-envelope thought experiment is appropriate in the context of our work. We believe that the calculation is discussed with appropriate qualifiers, but we will now more clearly state those considerations in the revised version.**

Page 11, line 12: see earlier comment on soil water sampling

**See response to general comment above.**

---

## Author Response (AR2)

Dear Dr. Weiler,

We could have been more clear to express that evaporation effects *on fractionation* were not considered in Brinkmann et al 2018. As a co-author of that study, I'm sure you considered the consequences of evaporation on fractionation. While we could go into a deeper comparison with Brinkmann et al., 2018, we can see that this is probably not in our best interest. As you remark, such comparisons are inhibited by our data being from a single snapshot, which conveys different meaning than other temporally focused datasets. Thus, we revised the comparison to which you object. There is also no need to step through every scenario that might affect SOI values because the metric is highly transparent: anyone can intuitively rationalize these effects. By listing all conceivable drivers of SOI variation, we would overwhelm and distract readers. Furthermore, lower-than-average precipitation in spring might affect the SOI, but if so, it would only be because it affected the contribution of summer versus winter precipitation in xylem, which is exactly what we are interested in inferring. Also, we explicitly state the challenges in interpreting intermediate SOI values "Samples with SOI values near zero approximate the annual average precipitation, and can potentially represent many possible mixtures of waters from spring, summer, autumn, and winter." (P3L13). We have made some changes: on page 10, where the comparison to Brinkmann et al occurs, we now (again) remind readers that our study differs from previous studies because it is a snapshot, and we also changed the comparison with Brinkmann et al 2018 by remarking that it provides temporal information that we lack; see tracked changes document.

Your second request is infeasible. While it is unclear whether you are asking us to calculate SOI on a time series of data at our sites, or on a time-series of data from other studies (e.g., Brinkmann et al), we do not have the necessary data to perform that calculation. This has been the case since our initial submission, so it seems highly unusual for this to be requested now. No changes are made.

We hope that you are satisfied with these changes.

Sincerely,

Scott Allen

[revised manuscript text omitted]